# Wide-Spectrum Antireflective Properties of Germanium by Femtosecond Laser Raster-Type In Situ Repetitive Direct Writing Technique

**Kaixuan Wang, Yubin Zhang** **, Jun Chen, Qingzhi Li, Feng Tang** , **Xin Ye \*** **and Wanguo Zheng \***

Research Centre of Laser Fusion, China Academy of Engineering Physics, Mianyang 621900, China; wangkaixuan20@gscaep.ac.cn (K.W.); zhangyub@mail.ustc.edu.cn (Y.Z.); chenjun19950110@163.com (J.C.); dearlqz@163.com (Q.L.); tangfengf3@caep.cn (F.T.)
\* Correspondence: xyecaep@mail.ustc.edu.cn (X.Y.); wgzheng_caep@sina.com (W.Z.)

**Abstract:** A femtosecond laser raster-type in situ repetitive direct writing technique was used for the fabrication of anti-reflective microhole structures in Germanium (Ge) in the visible near-infrared range (300–1800 nm). This technique builds a layer of microstructured arrays on the surface of Ge, enabling Ge to exhibit excellent anti-reflective properties. The large-area micro-nanostructures of Ge were fabricated using femtosecond laser raster-type in situ repetitive direct writing. Ge microstructures are characterized by their structural regularity, high processing efficiency, high reproducibility, and excellent anti-reflective properties. Experimental test results showed that the average reflectance of the Ge microporous structure surface in the range of 300–1800 nm was 2.25% (the average reflectance of flat Ge was 41.5%), and the lowest reflectance was ~1.6%. This microstructure fabrication drastically reduced the optical loss of Ge, thus enhancing the photothermal utilization of Ge. The many nanoburrs and voids in the Ge microporous structure provided excellent hydrophobicity, with a hydrophobicity angle of up to $133 \pm 2°$ (the hydrophobicity angle of flat Ge was $70 \pm 2°$). The high hydrophobicity angle allows for strong and effective self-cleaning performance. The femtosecond laser raster-type in situ repeatable direct writing technology has many desirable properties, including simplicity, high accuracy, flexibility, and repeatability, that make it one of the preferred choices for advanced manufacturing. The Ge micro-nanostructured arrays with excellent optical anti-reflective properties and hydrophobicity have become an attractive alternative to the current photo-thermal absorbers. It is expected to be used in many applications such as solar panels, photovoltaic sensors, and other optoelectronic devices.

**Keywords:** femtosecond laser processing technology; anti-reflection; self-cleaning; microholes



## 1. Introduction

The global energy problem has been one of the more intractable issues in recent decades, leading the world to turn to renewable sources of energy [1]. Sunlight is the most abundant and accessible source of renewable energy. The efficient use of solar energy can be achieved by increasing the reflectivity of materials in the broadband visible near-infrared range [2]. Currently, the main method of improving the anti-reflective properties of materials is to apply reflectance-reducing coatings to the surface. Compared with other semiconductor materials such as Si and GaAs, Ge has a low coefficient of expansion, high temperature and pressure resistance, chemical stability, and high overload resistance. Ge also has good semiconductor industrial compatibility, as well as good narrow band gap and infrared transmission [3–5]. Thus, Ge is widely used in micro- and nano-device fabrication and related industry fields [6,7]. Polished Ge surfaces have high reflectivity, making it difficult to perform effective light trapping. This limits its optoelectronic conversion efficiency in photonic devices. Therefore, we constructed novel micro-nanostructures on the surface of Ge to enable it to achieve low reflectivity in the broadband spectral range. Conventional

methods for the fabrication of micro-nanostructures include photolithography, nanoimprinting, and electron beams. However, these routes are expensive and time-consuming, and are not suitable for the processing of large-area micro-nanostructures. In contrast, femtosecond laser processing technology is widely used for the fabrication of micro-nanostructures due to its low cost, flexibility, non-contact methodology, diversity of processed materials, and other advantages. Femtosecond lasers have a very short pulse width ($10^{-15}$ s), peak power (power density after focusing up to $10^{22}$ w/cm$^2$), and a small heat affected zone (HAZ), characteristics which enable non-damaging ablation of materials [8]. It can be used for the micromachining of brittle materials with high hardness. Additionally, femtosecond laser processing technology is able to efficiently fabricate patterned surfaces such as microporous structures, inverted pyramids, micro- and nano-cylindrical structures, and micrometer slot array structures on the centimeter scale [9,10]. However, the femtosecond laser processing of microstructures still has some drawbacks, such as debris deposition and large surface roughness after processing [11,12]. Wet etching techniques supplementing femtosecond laser processing can mitigate these difficulties and improve processing efficiency [13,14].

Li's group used femtosecond laser pulses to prepare subwavelength micro/nanostructure arrays on Ge surfaces. The evolution of multi-scale surface morphology from periodic micro/nanostructures to V-shaped microgrooves was achieved [15]. Nayak's group used femtosecond laser pulses to process germanium in an SF6 environment. Regular arrays of nanospikes were obtained by modulating the laser energy density and the number of pulses [16]. Wang's group induced large well-aligned Ge nano-island arrays using femtosecond laser (120 fs, 800 nm, 1 kHz). Field emission (FE) measurements showed that the Ge nano-island arrays had high emission current density and good stability [17]. Zhao's team also studied how different environments (special gas and liquid environments) can affect laser processing results, which in turn affect the etching results. This enables efficient and high-quality processing of Ge materials [18].

In the process of laser interaction with Ge materials, there may be some side effects that endanger the performance of the material. These include the heat affected zone (HAZ), micro-cracks, point defects, etc. [19–21]. These key factors are worth considering in order to achieve high precision ablation of the material and quality control. Changing the surface wetting characteristics of Ge is also critical in some applications. For example, in haze and sandstorm environments, dust particles are deposited on the surface of antireflective devices. Then, in times of high humidity, these particles absorb water and become wet. In this situation, a Ge surface with self-cleaning properties would demonstrate its value. In this paper, the femtosecond laser raster-type in situ repetitive direct writing technique is used to construct composite micro-nanostructures with microholes and nanoscale burr bumps on Ge surface, which can significantly improve its anti-reflection hydrophobic performance [22–24].

In this study, microhole structures were prepared on the Ge surface via orthogonal machining using femtosecond laser raster-type in situ repetitive direct writing technology. Microporous structures were obtained after cleaning the surface debris with the aid of 10% HF-assisted chemical etching. This paper mainly regulates the period uniformity of Ge surface structure by adjusting laser irradiation parameters, processing methods, and cleaning methods [25,26]. We also investigated the mechanism of microstructure formation, optical properties (spectral broadband antireflection) and wettability [27–30]. We believe that a periodic dense microholes structure on Ge surface is the key to improving the anti-reflection performance. Ge microholes allow for better optical impedance between air and microholes. The densely arranged microholes structure on Ge increases the in-plane scattering of incident light and reduces the overall reflectivity, further suppressing reflections and capturing sunlight. The burr and void structures around the microholes can achieve super-hydrophobic properties. The surface of the super-hydrophobic Ge composite micro-nano structure can establish self-cleaning, which can reduce the influence of pollutant enrichment and water droplets [31,32]. In this work, experiments were carried out to prepare patterned Ge composite micro-nanostructures with good uniformity, repeatability,

and a large area on Ge surface using the femtosecond laser in situ repetitive direct writing technique. The results demonstrate that Ge microholes structures can effectively reduce reflection, and the average reflectivity of Ge microporous structures is less than 2.25% in the range of 300–1800 nm. The Ge composite microstructures also have good hydrophobicity properties. Overall, the anti-reflective properties of Ge microstructures are expected to play an important role in future photonics, wireless communications, and optical semiconductor devices [33,34].

## 2. Experimental

The Ge used in the experiment had <111> crystal direction, was N-type, was 20 mm × 20 mm × 1 mm in size, and the purity was 99.9999%. First, the Ge surface was cleaned using deionized water, acetone (AR ≥ 99.5%), and isopropyl alcohol (IPA, AR ≥ 99.5%) placed into each ultrasonic for 15 min. Finally, it was washed with deionized water and dried. The experimental operation was carried out in a clean room. Then, femtosecond laser direct writing was carried out in the atmospheric environment. The pulse of femtosecond laser used for the experiment was 130 fs, the frequency was 1 KHz with 10×, NA = 0.25 objective and 20×, NA = 0.4 objective. The detailed operation process was as follows: A cleaned Ge sample was selected and put into the 3D displacement stage. The laser power was set to 1.0–20 mW, the speed of the displacement table was set to 3000 μm/s, the period was set at 10 and 15 μm, and then the number of direct writing repetitions (three, five, seven, or 10 times) was set for laser direct writing. Finally, different Ge microholes structures were obtained after 5 min of ultrasonic cleaning with 10% HF mass concentration. Contact between HF acid and the human body was forbidden, so protective equipment was worn and extra precautions were taken. Then, the surface properties of the sample were tested. The surface morphology of the sample was analyzed by scanning electron microscope (SEM), and the reflectance data of the samples were measured and collected by UV-VIS-NIR spectrophotometer in the band range of 300~1800 nm. The static water contact angle of the sample was measured by a contact angle tester. The volume of the water drop was 2.5 μL, and the measurement error was ±2°.

## 3. Results and Discussion

Figure 1 shows the route of femtosecond laser raster-type in situ repetitive direct writing technology and wet cleaning process to fabricate a Ge microholes structure. Ge is one of the most important materials in the integrated circuit industry; therefore, experimental fabrication of micro-nanostructured surfaces with functional Ge was a good way to evaluate a new fabrication technique. Firstly, the femtosecond laser beam was focused on the Ge surface using a focusing objective. The wavelength of the femtosecond laser was 800 nm with a pulse width of 130 fs, a repetition frequency of 1 kHz, and an energy fluence of 20–50 J/cm$^2$. Microhole arrays were formed on the Ge surface after repeated ablation by the femtosecond laser (three, five, seven, or 10 times).

A large amount of debris existed on the Ge surface after direct writing, as shown in Figure 2A. Finally, the Ge microholes structure was obtained using ultrasonic assisted cleaning with HF (10% wt) for 5 min. The etching process was divided into two stages. First, rapid etching removal was performed in the initial femtosecond laser raster-type repetitive direct writing stage, which produced a large amount of debris and some melting layers and modified zones. The direct writing process was carried out in air, and the resulting surface debris became partially transformed into GeO$_2$. Then, the HF acid application removed debris and partially modified regions under sonication. The nanostructure of the microholes was altered with the etching of HF acid.

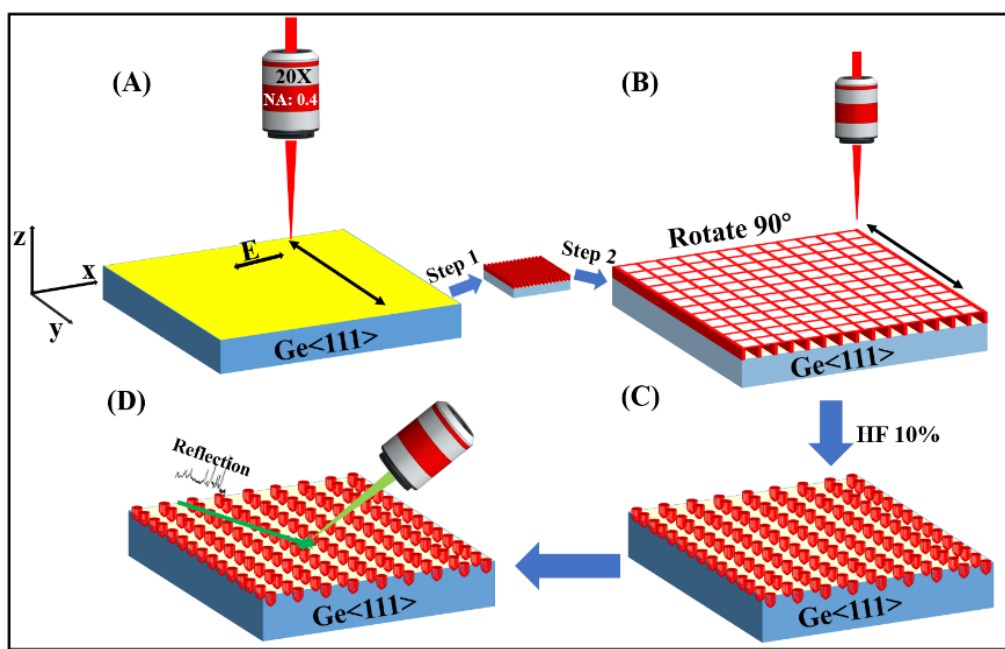

**Figure 1.** Schematic diagram of femtosecond laser raster-type in situ repetitive direct writing technique. (**A**) Schematic of the first direct writing with a femtosecond laser, (**B**) The second direct writing is perpendicular to the direction of the first direct writing, (**C**) Wet etching with HF, (**D**) Optical performance test.

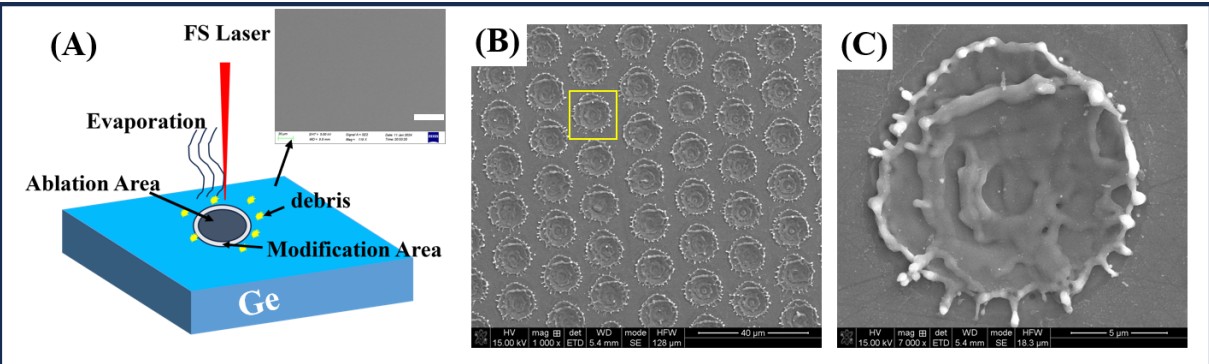

**Figure 2.** (**A**) Schematic diagram of mono-pulse femtosecond laser ablation (in the illustration is a SEM image of Ge before ablation, Scale bar: 40 μm), (**B**,**C**) SEM diagram of Ge microstructure prepared (objective lens 10×, NA 0.25, direct writing speed 12,000 μm/s, laser energy 40 J/cm$^2$).

### 3.1. Ablative Morphology of Mono-Pulse Femtosecond Laser

The surface morphology of Ge under single pulse femtosecond laser was studied in this section. Studies were performed according to the formula: $N = 2 W_0 f/V$, where $W_0$ was the size of the beam spot (2.44~3.9 μm), f was the laser frequency, and V was the direct write speed. The surface of Ge under single pulse was obtained in this section, as shown in Figure 2, in which the objective lens was 10×, NA = 0.25, and the direct writing speed was 12,000 μm/s with 40 J/cm$^2$ of laser energy. As shown in Figure 2B,C, aggregates of fine microstructures, including microcavities and nanoparticles, were observed near the edges of femtosecond laser action. These nanoparticles and the recast layer were fused to each other and onto the Ge surface. Debris and heat affected zones (HAZ) occurred at the edges of the ablation holes because they were ablated in an air environment. At high energy flux density, the ablation mechanism of the femtosecond laser was mainly phase explosion. Droplets violently erupted from the surface during ablation and were deposited and solidified at the hole edges. The formation of sputtering raised crowns of remelts led to

a degradation in the structural quality. The first thing that happened when a femtosecond laser was applied to a Ge material was the deposition of the laser energy into the material. The energy distribution in the temporal and spatial domains of the femtosecond laser during ablation determined the morphology distribution. Due to the ultra-fast and ultra-intense nature of femtosecond lasers, a strong non-linear absorption effect occurred when focused into Ge materials. When high energy was injected into Ge material, it caused the Ge lattice temperature to exceed its melting temperature, and thus the surface of the material underwent a rapid phase change (melting or gasification). A portion of the energy was converted into the kinetic energy of the lattice ions undergoing ablation, causing individual atoms, ions, molecules, or clusters to leave the surface. Finally, solidification occurred when the temperature dropped below the melting point. At this point, the single-crystal Ge of the ejecta was converted to polycrystalline or amorphous Ge (nano-burr in Figure 2C). The experimentally fabricated microholes had uniform morphology, and were produced in two steps. First, micron pits and nanoscale burrs were produced by focusing femtosecond laser irradiation on the Ge surface, causing it to undergo absorption, melting, ablation, and surface re-consolidation, during which amorphous Ge was produced. Then, the laser ablated sample was then placed in HF solution for chemical etch cleaning, which quickly removes debris and some oxides. In this process, the etching rate of amorphous Ge was much faster than that of crystalline Ge. The experimental fabrication of Ge microporous structures was carried out in air. A portion of the ejected Ge rapidly combined with oxygen to form germanium oxide and was removed by HF acid, while most of the fused cast layer recrystallized without being removed. Factors affecting the etching rate of Ge in HF include, for example, ambient temperature and solution concentration. The etching rate was also closely related to the dangling bond density and energy level of Ge. Under the same etching environmental conditions, the etching rate of amorphous Ge was high because the average dangling bond density and back-bonding energy level of crystalline Ge atoms were higher than that of amorphous Ge. Thus, the amorphous Ge exposed to the etching solution was removed quickly. The experimentally formed micropores had good homogeneity and were surrounded by remelts around each microhole.

In this section, the morphology of the Ge surface at different femtosecond laser energies was again investigated in detail in a comparative manner. The experimental conditions were an objective lens of $10\times$, NA = 0.25, a direct writing speed of 12,000 µm/s, and a period of 15 µm. As can be seen in Figure 3A, under high-energy single-pulse (energy of 60 J/cm$^2$) ablation, the ejecta of Ge material deposits and solidifies to form burr-bumped remelts at the edges of the microholes and forms very large microholes (15 µm in diameter). There was a distinct bulge in the middle of the microholes. This was formed by the high energy generated by the femtoseconds, which caused a large amount of thermally melted Ge to form in this region during the cooling process due to the high-pressure reflux effect. During the experiment, we also discovered that there was another portion of Ge that was removed in the form of smoke. After that, by attenuating the energy through the attenuating sheet, we found that when the energy is about 30 J/cm$^2$, as in Figure 3B, the microhole structures formed have the largest number of nanoburrs, and the heat affected areas are relatively small. More nanoburrs result in better performance in terms of hydrophobicity. When the energy was reduced to 10 J/cm$^2$, the microporous structure formed was the smallest. There were only a few nanoburrs around the micropores, which was not conducive to good hydrophobic performance (Figure 3C).

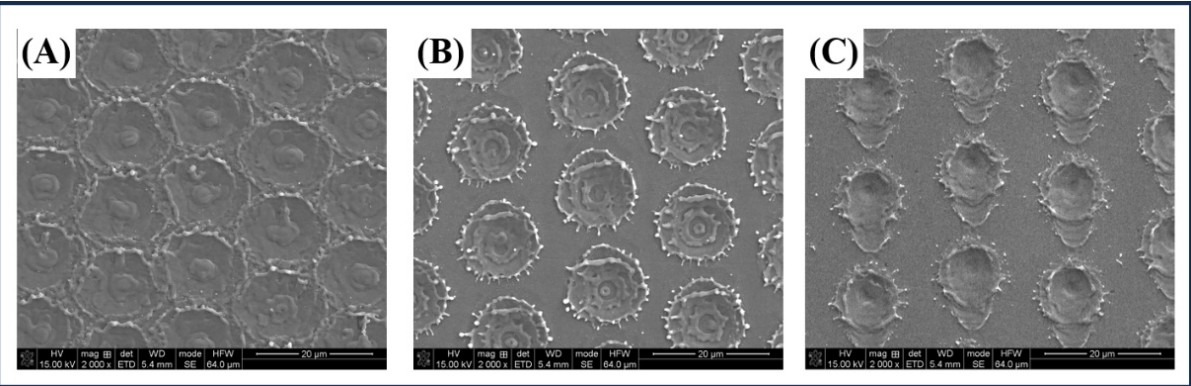

**Figure 3.** SEM morphology of Ge surface at different pulse energies. (**A**) 60 J/cm$^2$, (**B**) 30 J/cm$^2$, (**C**) 10 J/cm$^2$.

### 3.2. Raster-Type In Situ Repeat Direct Writing Technology

Figure 4A–D were fabricated by laser raster-type in situ repetitive direct writing techniques, carried out three, five, seven, and 10 times. The experimental parameters were as follows: the objective lens was 20×, NA = 0.4, the period was 10 μm, the writing speed was 3000 μm/s, and the laser energy was 25 J/cm$^2$. From Figure 4A–D above, it could be seen that at a relatively low number of direct writes, more burrs were produced, and the depth of the grating was shallower. As the number of direct writings increased, it did not reduce the burr, but deepened the depth of the groove. From the yellow ring in Figure 4B–D, we can see the craters ablated by the femtosecond beam spot, which is the result of the repetitive action of multiple pulses of femtosecond laser. An increase of the number of pulses produced many more eruptions, and the resulting deposited materials gradually became more and more abundant, to the point of occasionally plugging holes. Figure 4E–H shows Ge surfaces fabricated using the following experimental parameters: objective lens 10×, NA = 0.25, period of 15 μm, writing speed 3000 μm/s, and laser energy of 40 J/cm$^2$, executed in a number of direct writings (three, five, seven, and 10 times). From Figure 4E, it can be seen that after three direct writes, there were many burrs and the depth of the grating was shallower, with only shallow circular pits. This was due to the presence of multiple pulses of the raster-type in situ repetitive direct writing technique. Subsequent pulses of high energy acted on the liquid Ge heated and melted by previous pulses. The ablation pits appeared as circular ripples, leading to the formation of rippled remelted material after their sputtering solidification, as shown by the yellow circles in Figure 4E. When the number of in situ repetitive direct writing technique applications was increased to five and then seven times, the high-energy pulse ablation resulted in a large reduction of burrs on the surface of the composite micro-nanostructures, generating even deeper grating grooves. There were some relatively deep micro-pits in the grating grooves (yellow circles in Figure 4F,G) and some nano-bumps and pores in the grating ridges. When the number of direct writes was increased to 10 times, deeper grating grooves and even deeper holes were formed (yellow circles in Figure 4H). Based on the above results, we considered that the ablation process of high energy flow density femtosecond pulsed laser irradiated Ge samples causing rapid heating and melting on the surface. This interaction and gradual expansion of the material surface was accompanied by the ejection of droplets or vapor-liquid mixtures in the process of expansion. The intensity distribution of the femtosecond laser was Gaussian, and the distribution of the molten layer in the ablation region was not uniform. The molten liquid flowed and solidified from the center region towards the edges, while at the same time the ejecta was deposited and solidified at the edges. Under this effect, the edges of the ablation grating gradually formed more crown-like remelts. As the number of scans increased, the deepening of the hole depth and the redeposition of the ejecta gradually dominated the phenomenon. In the ablation mechanism of femtosecond laser pulses, the interaction of the subsequent pulse with the

ejecta was essential to consider. Because of the continuous expansion of the ejecta under the action of the previous pulse, the ejecta had already diffused beyond the laser irradiation range. The subsequent pulses only provided secondary ablation of nanoparticles within the irradiated region, but could not prevent diffusion and deposition of nanoparticles outside the irradiated region. The low magnification objective had a larger beam spot and higher energy, so there were few nano-bumps on the grating ridges under multiple direct writings.

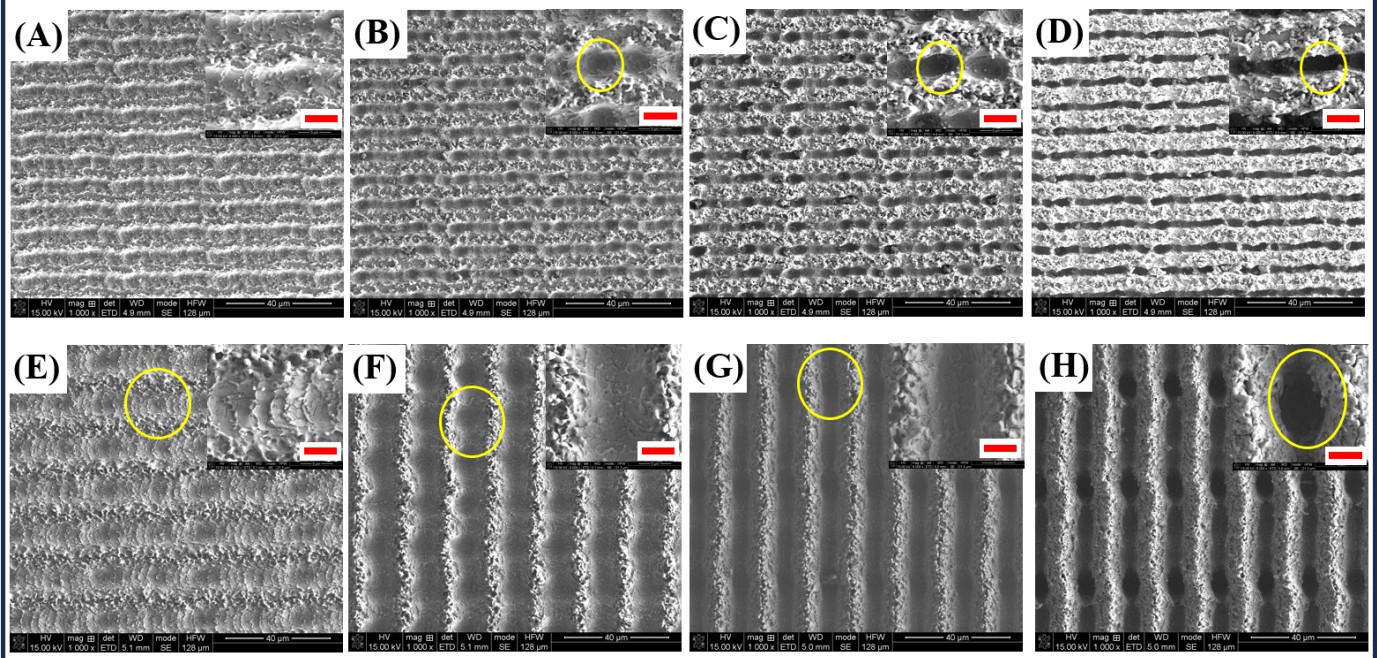

**Figure 4.** SEM images of femtosecond laser grating-based in situ repetitive direct writing. (**A–D**) different times 3, 5, 7, 10 under 20× objective, respectively; (**E–H**) different times 3, 5, 7, 10 under 10× objective, respectively. (The red scale bar is 5 μm in the inset image).

### 3.3. Orthogonal Laser Direct Writing

Figure 5A–D show the femtosecond laser raster-type in situ writing technique applied three, five, seven and 10 times. The objective lens was 20×, the period was 15 μm, the direct writing speed was 3000 μm/s, and the laser energy was 25 J/cm$^2$. Scanning was conducted perpendicular to the first direct writing direction of the fabrication, which is orthogonal to the approach. Figure 5A represents fabrication orthogonally after three repeats of direct writing. Due to the low number of raster-type in situ repetitions and the low repetition rate of the femtosecond beam spot, relatively few microholes, especially deep ones, were observed throughout the surface. Only sputtered burr-like nanostructures were present on both sides of the femtosecond laser scanning track. When the number of direct writing repeats was five or seven times, the number of microholes produced on the surface gradually became higher and their definition clearer, and the nanoburrs on the grating ridge were distributed uniformly. With an increase in the number of direct writes to 10 times, the periphery of the microholes became covered with nanoprotrusions and nanocavities. these nanoparticles were fused with each other onto the Ge surface, and the phenomenon of blocking holes appeared (Figure 5D, zoomed-in view). Because the femtosecond beam spot under the high-magnification objective was too small and the energy was insufficient, the microholes formed under several direct writings became deeper while the sprayed and deposited materials gradually became too much to be removed in time. Therefore, there were still some fused castings inside the microholes, and some blocking burrs appeared around the microholes.

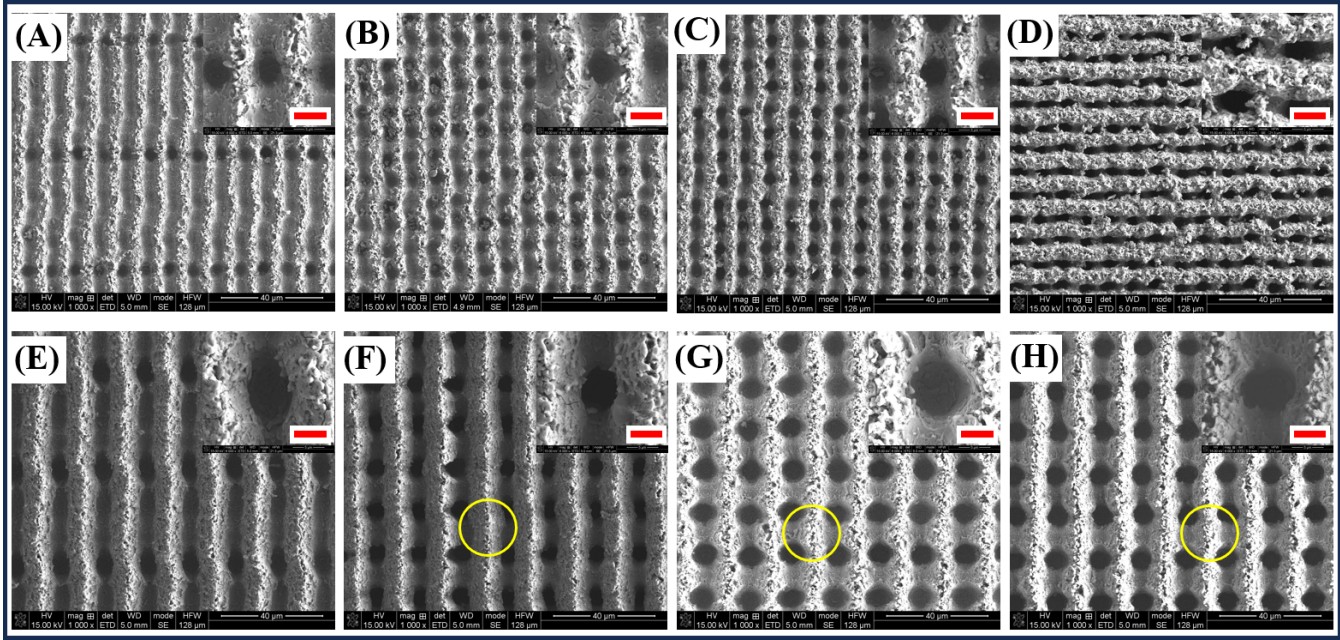

**Figure 5.** SEM images under orthogonal straight writing: (**A–D**) different times 3, 5, 7, 10 under 20× objective, respectively; (**E–H**) different times 3, 5, 7, 10 under 10× objective, respectively. (The red scale bar is 5 μm in the inset image).

The surface of a Ge structure fabricated using 10× objective lens, laser energy of 40 J/cm$^2$, and a laser scanning speed of 3000 μm/s is seen in Figure 5E. When the number of orthogonal writes was three, the distribution of micropores was not homogeneous. As shown in Figure 5F, when the number of direct writings was five, the microvia formed clearly at the bottom of the grating groove. This was due to the orthogonal, multiple direct-write approach that allowed the femtosecond beam spot to act on the same aperture location for a long period of time. The material inside was ablated out and transferred to the periphery: the sides of the microholes were solidified by the deposition of molten material ejected from the repeated scans to form grid ridges. The yellow circle in Figure 5F shows the highest grid ridge, as it is almost free of direct femtosecond laser beam spot irradiation. In the femtosecond laser-treated region, not only microhole shaped structures appeared; a large number of nano-burr structures and voids were also created on the grating ridges. This was a recast layer formed by multiple ejections and depositions of molten Ge material. When the number of orthogonal direct writes reached seven and 10 times, stable and uniform fine microstructures, including microcavities and aggregates of nanoparticles, were produced, as shown in Figure 5G,H. This phenomenon was related to the thermal effect of the Gaussian beam. As the laser power increased, both the energy and pressure increase, and the irradiated area melts the solids in the central area into liquid sprays, and a portion evaporates directly. The area around the ablation crater continued to be subjected to thermal effects through repetitive melt annealing to form a fused layer.

*3.4. Optical and Hydrophilic Properties*

After the Ge samples treated in an orthogonal manner by femtosecond laser raster-type in situ repeat direct writing technique were ultrasonically etched in 10% wt HF solution for 5 min, the treated areas showed as pitch black. This indicated a significant change in the optical properties of the Ge samples. Ge had zero transmittance in the visible to near-infrared range. Figure 6A shows the reflectance spectra of Ge micro-nanostructures fabricated under a 20× objective, which decreased with the number of direct writes. The average reflectance in the 300–1800 nm band was 3.4% (10 writes). Figure 6B shows the reflectance spectra of Ge micro-nanostructures fabricated under a 10× objective lens. The

reflectance decreased as the number of direct writes increased, with the lowest average reflectance of 2.24% (7 times) in the 300–1800 nm band, and an average emissivity in flat Ge samples of 41.5%. From Figure 6, it is seen that Ge micro-nano structures exhibited low reflectivity compared to flat Ge (green line) in both 300–1800 nm bands, and the reflectivity was drastically reduced. The Ge micro-nanostructures fabricated using femtosecond laser raster-type in situ repetitive direct writing technology formed a large number of deep microholes and nano-voids, which increased the geometric reflection of light and enhanced the light-trapping effect to further reduce reflectivity. Deeper microholes and high porosity as well as processing efficiency were obtained with a low magnification objective and higher laser energy. The Ge micro-nanostructures fabricated using the laser raster-type in situ repetitive direct writing technique and HF wet etching have low reflection efficiency, which gives them better photothermal absorption performance. This is expected to drive good development in the field of optoelectronic devices and infrared optoelectronics. The micro-nanostructured Ge surface retained nanoburrs and pores, which also contributed to enhanced hydrophobicity.

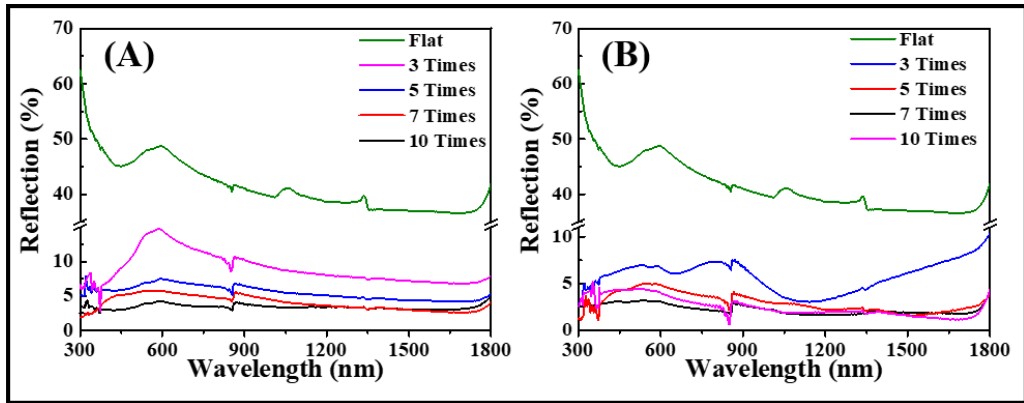

**Figure 6.** Reflectance curves of Ge for femtosecond laser raster-type in situ repetitive direct writing. (**A**) prepared by direct writing with 20× objective lens; (**B**) prepared by direct writing with 10× objective lens.

Femtosecond laser ablation was shown to be a powerful tool for controlling surface wettability. In this paper, composite micro-nanostructures with hierarchical surfaces were fabricated on the surface of Ge wafers using the femtosecond laser raster-type in situ repetitive direct writing technique. From Figures 5 and 6, we know that a periodic hierarchical array of microholes (microhole period was 15 μm, diameter was 10 μm) was formed on the Ge surface. The periphery of each of these Ge micropores is further wrapped and covered by an abundance of burrs and pores of a few hundred nanometers or so. The rough micro-nanostructures of the graded type after laser ablation increase the hydrophobicity. Figure 7 shows the water contact angle on the Ge surface after femtosecond laser ablation. We found that the microstructure water contact angle was improved by multiple direct writings with a 20× high magnification objective (direct writings seven times, CA~102 ± 2°, in Figure 7B; direct writings 10 times, CA~125 ± 2°, in Figure 7C), but it was still smaller than that of the direct writings with 10× low magnification objective (direct writings seven times, CA~133 ± 2°, in Figure 7D; direct writings 10 times, CA~131 ± 2°, in Figure 7E). The flat Ge water contact angle was 70 ± 2°. As a result, femtosecond laser ablation of Ge surface micro-nanostructures could exhibit excellent super-hydrophobicity and achieve a self-cleaning effect. This broadens outdoor application scenarios for Ge micro-nanostructures, which can be widely used in solar photothermal devices.

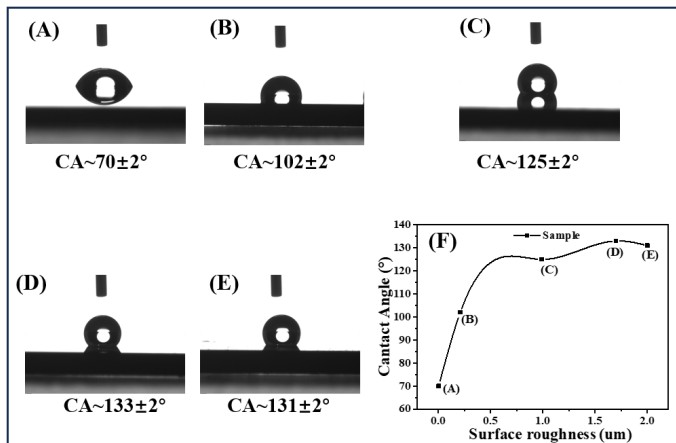

**Figure 7.** Water contact angle (CA) images: (**A**) flat Ge; (**B**,**C**) 20× objective lens direct writing 7 times and 10 times; (**D**,**E**) 10× objective lens direct writing 7 times and 10 times; (**F**) The contact angle values as a function of surface roughness.

## 4. Conclusions

In this paper, a femtosecond laser raster-type in situ repetitive direct writing technique is shown to provide a simple and efficient alternative for fabricating high-performance Ge microhole structures. Ge microhole structures have a strong anti-reflective effect in the visible-near-infrared range because of the deep microhole structures, which efficiently scatter, trap, and absorb visible light. The Ge microhole structures and the large number of nanoburrs and void structures around them also enhance the hydrophobic properties. In this experiment, the femtosecond laser raster-type in situ repetitive direct writing technique successfully fabricates Ge microhole structures with good uniformity, high precision, and a large area. We analyzed the mechanism of microhole formation in detail. In optical test experiments, microhole structures exhibited excellent anti-reflective properties with an average reflectivity as low as 2.25% in the 300 to 1800 nm band; they also exhibited strong hydrophobic properties, with a water contact angle of $133 \pm 2°$. In general, Ge-based broadband anti-reflective microstructures based on femtosecond laser raster-type in situ repetitive direct-write process technology can significantly reduce the reflectivity of Ge in the visible to near-infrared range and improve self-cleaning performance. Therefore, the experimental fabrication of self-cleaning and reflection-reducing Ge micro-nanostructures can enhance the photothermal absorption and high photovoltaic conversion efficiency of micro-nano devices. This knowledge is expected to be applied in many fields such as photonics, wireless communication, and optical semiconductor devices.

**Author Contributions:** K.W.: Conceptualization, Formal analysis, Investigation, Data curation, Writing—original draft, Writing—review & editing. Y.Z.: Conceptualization, Formal analysis, Investigation, Data curation. J.C.: Conceptualization, Formal analysis, Investigation. F.T.: Conceptualization, Formal analysis, Revision. X.Y.: Conceptualization, Formal analysis, Revision. X.Y. and Q.L.: Formal analysis, Funding acquisition. W.Z.: Conceptualization, Formal analysis, Revision. All authors have read and agreed to the published version of the manuscript.

**Funding:** This research was funded by National Natural Science Foundation of China (No. 61705204). The Open Project Program of Key Laboratory for Cross-Scale Micro and Nano Manufacturing, Ministry of Education, Changchun University of Science and Technology (CMNM-KF 202110).

**Institutional Review Board Statement:** Not applicable.

**Informed Consent Statement:** Not applicable.

**Data Availability Statement:** The data provided in this study may be obtained from the corresponding author upon reasonable request.

**Conflicts of Interest:** The authors declare that they have no known competing financial interests or personal relationships that could have appeared to influence the work reported in this paper.

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
