# Peer review of "Wide-Spectrum Antireflective Properties of Germanium by Femtosecond Laser Raster-Type In Situ Repetitive Direct Writing Technique"

_coatings, doi:10.3390/coatings14030262_

Round 1
Reviewer 1 Report
Comments and Suggestions for Authors
Dear Authors,
The manuscript is interesting and describes the use of germanium foil as a surface with antireflective properties.
Some suggestions should be considered:
i) Replace the title Ge with Germanium;
ii) Review the keywords - Ge is not a keyword but a symbol of the chemical element;
iii) In the abstract the Ge symbol appears 10 times, and there is no word Germanium.
iv) Review the manuscript regarding indexing of references; and shorten the first paragraph of the introduction.
v) The discussion of the results revolves around images of the surface after laser ablation. Could include an image before ablation.
vi) Present an EDS spectrum showing the composition.
Adjust the formatting (text, captions, figures, etc.) according to the Coatings template.

Reviewer 2 Report
Comments and Suggestions for Authors
In this work, the authors prepared and characterized microporous composite nanostructured array structures on Ge to achieve surfaces with anti-reflective properties using a femtosecond laser raster-type in-situ repetitive direct writing technique. These surfaces can be applied to different devices. The manuscript is clear and detailed, the experiments and the results are well described, and the conclusions follow the results, therefore, it can be considered for publication if the following revision is considered:
- Please indicate in the Experimental Section the precautions that were taken into account when handling HF during the ultrasonic cleaning.
- In section 3.2, the authors wrote “The objective lens is 10X NA 0.25, the period is 15um, the writing speed is 3000um/s, and the laser energy is 40J/cm2.” Please verify the units of 15um and 3000um/s. Please verify the units in the whole article.
- Please replace “~102°±2°” by “~102±2°”. The same should be done for similar cases throughout the document.
- Why did the authors not represent the contact angle values as a function of surface roughness? Please include some information about this in the manuscript.
Reviewer 3 Report
Comments and Suggestions for Authors
With regards to the structure of the manuscript:
l Motivation in the introduction mention Ge has a high absorption coefficient but then mentions it has high reflectivity? Contradictory? (Page 2)
l Author mention there is a need to endow Ge with “better” optical properties and lists some applications. But what specific optical property is desired? Lower reflectance, can infer increased transmittance or increased scattering, very unclear what the aim is.
l Authors mention self-cleaning with regards to haze and dust storms and debris precipitating. So self-cleaning with regards to water (hydrophobicity) seems to not relate with particle cleaning? (Page 3)
l References regarding the femtosecond laser processing of Ge seem to be not well organized in terms of relevance to the current study. i.e., Works regarding pure ablation are mentioned with notable length but seem to be not related at all as this work is about surface texturing and not cutting.
With regards to the experimental procedures:
l Analyzing the reflective properties solely with UV-VIS-NIR spectroscopy is insufficient, as it cannot determine if a change in transmittance is due to reflectance, scattering, or absorption. Considering the micro and nanostructures formed in this study, the change in “reflectance” may simply be due to an increase in scattering, which is typically not ideal for optoelectrical applications.
l Authors discuss the femtosecond laser material interactions in terms of melting, boiling, and evaporation (mostly in terms of lattice physics). Considering the repetition rate of the laser system, this seems to be a severely flawed way of discussing the mechanism for structural formation.
l Authors mention that the chemical etching done afterwards should remove oxide layers and amorphous structures formed due to laser ablation. Nonetheless, the structures consist of many bumps. Do the authors think these bumps are crystalline Ge then? More material analyses on the surface are required. Relating to this, hydrophobicity is commonly attributed to surface functional groups, but this detail seems to be entirely lacking.
Comments on the Quality of English LanguageExtensive revision is recommended.
Reviewer 4 Report
Comments and Suggestions for Authors
By studying the figures of the manuscript and its captions, we find the results of an interesting and fruitful work. However, when reading the text of the work, we encounter serious interpretation difficulties.We believe that the English language of this study is occasionally quite hard to understand. This is due to the fact that most of the sentences are far too long, over-complicated, contain repetition of words and inconsistencies.We suggest using shorter sentences with clear structures. I consider it also necessary to standardize the number of references in the text, according to the general editing rules.
Comments on the Quality of English Language
We believe that the English language of this study is occasionally quite hard to understand. This is due to the fact that most of the sentences are far too long, over-complicated, contain repetition of words and inconsistencies.We suggest using shorter sentences with clear structures. I consider it also necessary to standardize the number of references in the text, according to the general editing rules.
Round 2
Reviewer 3 Report
Comments and Suggestions for Authors
Thank you for the responses. They were mostly satisfactory for publication.
However, I have one additional question, as my original question may not have been understood clearly by the authors.
Particularly, question 6, I was questioning whether the authors think the modification process is a non-thermal process or a thermal process. Since the authors mention boiling and evaporation, I assumed they conclude it is thermal. But considering a low repetition rate, I am not sure if it is a thermally dominated process or a non-thermal optical process such as Coulomb explosion. Discussion on this is appreciated.
Comments on the Quality of English LanguageRe-checking the entire manuscript is recommended.
Reviewer 4 Report
Comments and Suggestions for Authors
I have the feeling that the authors' changes significantly improve the manuscript's comprehensibility and thus the quality of the work. After accepting the changes, I recommend the manuscript for publication.
